# Qigong Therapy for Stress Management: A Systematic Review of Randomized Controlled Trials

**DOI:** 10.3390/healthcare12232342

**Published:** 2024-11-23

**Authors:** Jung-Ho Oh, Soo-Hyun Sung, Jang-Kyung Park, Soobin Jang, Byung-Cheul Shin, Sangnam Lee

**Affiliations:** 1Department of Qigong, College of Korean Medicine, Daegu Haany University, Gyeongsan 38609, Republic of Korea; nature3681@gmail.com; 2Department of Policy Development, National Institute for Korean Medicine Development, Seoul 04516, Republic of Korea; koyote10010@nikom.or.kr; 3Department of Obstetrics and Gynecology of Korean Medicine, Pusan National University Korean Medicine Hospital, Yangsan 50612, Republic of Korea; vivat314@pusan.ac.kr; 4Department of Preventive Medicine, College of Korean Medicine, Daegu Haany University, Gyeongsan 38609, Republic of Korea; suebin@nate.com; 5Division of Clinical Medicine, School of Korean Medicine, Pusan National University, Yangsan 50612, Republic of Korea; 6Department of Korean Medicine Rehabilitation, Pusan National University Korean Medicine Hospital, Yangsan 50612, Republic of Korea

**Keywords:** Qigong, stress reduction, anxiety, depression, quality of life, meta-analysis

## Abstract

Background/objective: Althouth Qigong is commonly used to manage stress, anxiety, and depression, there have been no systematic reviews on Qigong therapy for stress management. This study aimed to analyze the clinical evidence of Qigong therapy for perceived stress. Methods: We conducted a systematic search for randomized controlled trials (RCTs) of Qigong using 11 electronic databases, namely MEDLINE, EMBASE, Cochrane Central Register of Controlled Trials, Physiotherapy Evidence Database, CINAHL, and Korean Medical databases (Korea Institute of Science and Technology Information, Korean traditional knowledge portal, KoreaMed, OASIS, RISS, and the National Library of Korea). We considered RCTs in which participants with perceived stress with no restrictions on age, gender, or ethnicity. Two reviewers independently assessed risk of bias of the included RCTs using the Cochrane risk of bias tool. Nine RCTs (China: *n* = 5, South Korea: *n* = 2, United States: *n* = 1, Brunei Darussalam: *n* = 1) were included in the systematic review. Results: The quality of the included trials was generally low, as only one was rated as high quality. For the primary outcome, a meta-analysis of two RCTs showed statistically significant results on the perceived stress scale comparing the Qigong therapy group with the no-treatment group (OR −0.60; 95% CI −1.02 to −0.17; *p* = 0.006). The results of three other studies showed that Qigong therapy had no significant effect compared to active-control therapy on the perceived stress scale outcome (OR −2.10; 95% CI −4.68 to 0.47; *p* = 0.11). Regarding the secondary outcomes, including depression, anxiety scale, and quality of life, the Qigong group showed statistical improvements in most studies; however, there was no difference in the pain scale between the two groups. In two studies, no adverse events occurred, whereas in one study, six cases (24% of participants) of mild muscle soreness were reported. Conclusions: This systematic review suggests the potential of Qigong therapy for stress management; however, it is difficult to draw specific conclusions. Future studies should standardize Qigong interventions and outcomes, establish sham control groups, and include larger sample sizes in RCTs.

## 1. Introduction

Qigong is a bioenergy therapy with a long history of use in many diseases [1]. Qigong is composed of two terms: qi meaning “energy flow”, and gong meaning “skill” or “achievement”. The main components of Qigong include training in consciousness, breathing, body movement, and the adjustment or stimulation of one’s own qi [2].

According to the Korean Standard Clinical Practice Guidelines for Korean Medicine developed by the Korean government and the Traditional Korean Medicine (TKM) Society, Qigong is recommended as a standard treatment method for mental health conditions, such as depression and anxiety disorders [3,4,5]. Additionally, the National Center for Complementary and Integrative Health (NCCIH) in the United States officially recognizes Qigong as complementary and alternative medicine. It is applicable not only to physical conditions such as fibromyalgia, chronic obstructive pulmonary disease (COPD), Parkinson’s disease, hypertension, and chronic heart failure, but also to mental health conditions such as substance abuse disorders, anxiety, and depression [5].

Looking at the clinical research trends related to Qigong, a systematic review of 14 clinical studies on Qigong therapy for cancer patients reported improvements in physical symptoms (fatigue, pain, numbness, dizziness), psychological symptoms (depression, sleep disorders), and quality of life (QoL) [6]. Additionally, a meta-analysis of 20 clinical studies involving 2349 participants on the treatment of hypertension with Qigong showed a statistically significant reduction in both systolic and diastolic blood pressure [7]. A systematic review of nine clinical studies on Qigong for managing type 2 diabetes reported statistically significant improvements in glycated hemoglobin (HbA1c), 2 h postprandial plasma glucose concentration (2hPG), insulin sensitivity, and blood viscosity [8]. Moreover, systematic reviews of Qigong have demonstrated statistically significant effects on heart disease [9], post-stroke management [10], and improvement in depression and anxiety symptoms [11].

The prevalence of stress has been reported to be between 22 and 28% [12,13], although, during the COVID-19 outbreak in 2019–2020, the global prevalence of stress was found to be 36.5% [14]. Stress in the form of anxiety, anger, depression, and confusion due to factors such as family, society, and other unpredictable elements (e.g., the spread of infectious diseases) can lead to mental health issues such as depression and suicidal ideation [15]. Additionally, when stress accumulates and reaches the exhaustion stage, where organ and tissue functions are diminished, physical illnesses such as hypertension, heart disease, gastrointestinal disorders, irritable bowel syndrome, headaches, respiratory diseases, and skin conditions can occur, which can negatively affect interpersonal relationships [16,17]. It is essential to manage stress to prevent it from progressing to serious stages such as mental illnesses, using methods like ‘time management, conflict resolution, communication skills, social support, humor, spirituality, meditation, exercise, yoga, and massage [18].

In a previous systematic review [19], eight randomized controlled trials (RCTs) on Qigong for stress and anxiety were reviewed in 2014. This study suggests that Qigong exercise relieved anxiety and reduced stress compared with wait-list controls. However, this review was conducted 10 years ago, and only three RCTs have specifically applied Qigong to stress. Therefore, the conclusions suggest positive effects on perceived stress, but the evidence is not strong enough.

To the best of our knowledge, there have been no published systematic reviews of Qigong for stress management. The aim of this study was to assess whether Qigong therapy can be effective in managing stress.

## 2. Materials and Methods

### 2.1. Types of Design

We performed a systematic review to investigate the evidence on the effectiveness and safety of Qigong therapy for stress management, following the recommendations of the Preferred Reporting Items for Systematic Reviews and Meta-Analyses (PRISMA) guidelines (2020) [20] (see Appendix A for the PRISMA checklist of systematic reviews). Our systematic review protocol was registered in an international prospective register of systematic reviews under the registration number PROSPERO 2024 (https://www.crd.york.ac.uk/prospero/display_record.php?ID=CRD42024581803, accessed on 31 August 2024).

The following studies were considered for inclusion: (1) full text articles, (2) English and Korean language papers, and (3) RCTs in clinical research. Exclusion criteria were (1) conference abstracts, letters, and comments, (2) papers published in languages other than English and Korean, (3) non-RCTs among clinical studies (e.g., case reports, case series, and case-controlled trials), (4) animal in vivo or in vitro experimental studies, and (5) non-Clinical trials (e.g., reviews, qualitative studies, surveys, and study protocols).

#### 2.1.1. Participants

We included studies in which participants perceived stress. We imposed no restrictions as to age, gender, or ethnicity.

#### 2.1.2. Types of Interventions

We considered for inclusion all studies assessing the efficacy of Qigong therapy for stress management. Qigong was considered to include movements, relaxation, exercises, meditation, and breathing conducted under the supervision of an experienced Qigong practitioner.

#### 2.1.3. Types of Comparisons

We did not set any restrictions on control group interventions. Therefore, control interventions included conventional treatments, complementary and alternative treatments, and no intervention.

#### 2.1.4. Types of Outcomes

We included studies that measured the effects of Qigong therapy on stress. We did not limit the types of stress scales used.

### 2.2. Data Sources and Searches

A search was conducted within eleven electronic databases: MEDLINE, EMBASE, Cochrane Central Register of Controlled Trials (CENTRAL), Physiotherapy Evidence Database (PEDRO), CINAHL, and Korean Medical databases (Korea Institute of Science and Technology Information, Korean traditional knowledge portal, KoreaMed, OASIS, RISS, and the National Library of Korea).

The search terms used were ((“Qigong” OR “Qi gong” OR “Qi-gong” OR “Chi gong” OR “Chi kung” OR “Chi chung” OR “Qi-training”) AND (“stress”) AND (“clinical trial”) OR (“randomized controlled trial” OR “randomized controlled trial”)). The search terms for each database are included in Appendix A

### 2.3. Study Selection and Data Extraction

Two authors (S.J. and J.-H.O.) independently screened the titles/abstracts of all the potential records. The articles were included and excluded according to the inclusion and exclusion criteria. Study characteristics of the included studies were summarized by PICO (P: patient, I: intervention, C: comparison, O: outcome). Two review authors (S.J. and J.-H.O.) independently extracted the data using a predefined data extraction form. Two independent reviewers (S.-H.S. and J.-K.P.) collected data on the authors’ information, sample size, interventions, outcome measures, main results, and adverse events. The primary outcome was the stress scale. Secondary outcomes included (1) psychological indicators including depression and anxiety, (2) quality of life, (3) health scale, (4) pain scale, (5), biomarkers, and (6) adverse events. Regarding Qigong interventions, we extracted Qigong type, details of the specific programs, program length, and frequency and duration of treatment.

All processes involved comparing results among researchers and reaching a consensus through discussion. Any discrepancies were resolved after consulting with the corresponding author (B.-C.S.) for final decisions.

Additionally, studies that included a structured Qigong program were incorporated. The Qigong type, program length, session length, frequency, and other details were not restricted. Cases where Qigong was practiced independently without the guidance of a practitioner or without a structured Qigong program were excluded from the research.

### 2.4. Assessment of Risk of Bias (ROB)

Two researchers (S.J. and J.-K.P.) independently evaluated the risk of bias using the Cochrane Collaboration’s ROB 2.0 tool [21]. ROB was divided into six domains: (1) randomization process, (2) deviations from intended interventions, (3) missing outcome data, (4) measurement of outcome, (5) selection of reported result, and (6) overall bias. For each domain, ROB was rated as low risk, high risk, or some concerns. Disagreements between researchers were discussed until consensus was reached.

### 2.5. Data Analyses

The RevMan 5.4 (version 5.4 for Windows; Nordic Cochrane Center, Copenhagen, Denmark) software was considered for statistical analyses. Continuous and dichotomous data are reported with 95% confidence intervals (CIs), and the continuous and dichotomous data are reported as differences or risk ratios, respectively. The I^2^ test was used to evaluate inter-study heterogeneity, with I^2^ values of 0–40%, 30–60%, 50–90%, and 75–100% representing the absence or mild, moderate, substantial, and full heterogeneity, respectively [22]. When a meta-analysis could not be performed because of considerable variation in the study characteristics, the results of the studies were qualitatively summarized.

## 3. Results

### 3.1. Study Selection and Description

The initial search by key words revealed 503 articles. After removing 271 duplicates, 232 papers were screened on their titles and abstracts, and 21 articles were identified to be read as full text. Of the remaining 21 articles, one article was excluded because it was only a protocol, five articles were excluded as case studies, two articles were excluded as case-controlled trials, two RCTs were excluded because they used mixed interventions in the experimental group, and two articles were excluded because they utilized education programs. Finally, nine papers [23,24,25,26,27,28,29,30,31] were included in this systematic review (Figure 1).

A total of 799 patients with stress were evaluated in nine RCTs, with 383 patients in the experimental group and 450 in the control group. Of these, four RCTs [27,29,30,31] were analyzed with meta-analysis. Of the nine RCTs, the majority were conducted in China (*n* = 5) [25,28,29,30,31], followed by South Korea (*n* = 2) [24,27]. One study was conducted in each of the United States [23] and in Brunei Darussalam [26]. Table 1 summarizes the RCTs, including the title, first author, year of publication, patients, sample size, interventions of the experimental and control groups, outcome measures, and main results.

### 3.2. Intervention

The types of Qigong therapy included Qigong exercises [23,26,28,29], the Brief Qigong-based Stress Reduction Program [24,27], Chan Mi Gong Qigong exercises [25], Yijinjing Qigong exercises [30], and Qigong fitness [31]. The specific programs used in the Qigong exercises for the four RCTs [23,26,28,29] were found to differ from each other. The Brief Qigong-Based Stress Reduction Program was the same across the studies. The details of the Qigong programs are presented in Appendix A

### 3.3. Effects of Qigong Therapy

#### 3.3.1. Primary Outcome

##### Stress Scale

Stress was significantly reduced in participants following Qigong therapy compared to no intervention [23,24,25,27,28,29], stretching training exercises (*p* < 0.05) [30], and cognitive-behavioral therapy (*p* < 0.05) [31].

As a result of the meta-analysis, in two studies [27,29], the Qigong therapy group showed statistically significant results on the perceived stress scale compared with the no-treatment group (OR −0.60; 95% CI −1.02 to −0.17; *p* = 0.006) (Figure 2a). In addition, three other studies [29,30,31] showed that Qigong therapy had no significant effect compared to active-control therapy on the perceived stress scale outcome (OR −2.10; 95% CI −4.68 to 0.47; *p* = 0.11) (Figure 2b).

#### 3.3.2. Secondary Outcomes

##### Depression Scale

Depression was reported to have a significant effect (*p* < 0.05) in four studies [26,28,29,30,31] compared to no intervention [28,29], stretching training exercises [30], and cognitive-behavioral therapy [31]. One study [26] comparing Qigong therapy to no intervention reported that the effect comparison between the two groups was not provided; however, the experimental group showed statistically significant improvements in pre- and post-intervention scores (*p* < 0.05), while the control group did not show improvements. One study [25] found no significant difference between the Qigong therapy and no-intervention groups, and another study [29] reported no significant difference between the Qigong therapy and Integrative Body–Mind–Spirit groups.

##### Anxiety Scale

In the group that received Qigong therapy, anxiety significantly decreased compared to the no-treatment group [24,25,27] and cognitive-behavioral therapy [31].

##### Quality of Life

In two studies comparing Qigong therapy to the no-treatment group [25,27], significant improvements were observed in all QoL subscales (*p* < 0.05). One study [24] reported significant improvements in some QoL items based on temperament types, while another study [23] found no differences between the two groups.

##### Health Scale

In one study [29] comparing Qigong therapy to no intervention, significant improvements were observed in physical symptoms (pain: *p* = 0.019, painless: *p* < 0.001), the Pittsburgh Sleep Quality Index (*p* = 0.023), and the Mind–Body Scale (Affliction: *p* < 0.001, Equanimity: *p* = 0.002). However, when Qigong therapy was compared to Integrative Body–Mind–Spirit, no significant differences were found in physical symptoms, the Pittsburgh Sleep Quality Index, or the Mind–Body Scale [29]. In a comparison between Qigong therapy and stretching training exercises, mental health scale scores improved (*p* < 0.001), while physical symptoms showed no improvement.

##### Pain Scale

When comparing Qigong therapy to no treatment [23] or stretching training exercises [30], no significant differences in pain were observed between the two groups.

##### Biomarkers

One study [25] found a significant difference in cortisol levels between the two groups (*p* < 0.001), while the other study [26] did not report a comparison of effects between the two groups; however, the experimental group showed statistically significant improvements in pre- and post-intervention values (*p* < 0.05), whereas the control group did not show improvements. One study [28] reported no significant differences between the groups in telomerase activity, plasma concentrations of interleukin-6, or tumor necrosis factor. In contrast, another study [29] demonstrated positive changes in plasma concentrations of interleukin-6 and interleukin-1β.

##### Adverse Events

Three studies [23,28,30] reported adverse events (AEs) of Qigong interventions, whereas the remaining six studies [24,25,26,27,29,31] did not report any AEs. Two studies [23,28] reported that no adverse events occurred. One study [30] reported 6 cases (24%) of mild muscle soreness in the Qigong group and 12 cases (48%) of mild muscle soreness in the stretching exercise group.

### 3.4. Assessment for ROB

The overall risk of bias for the included studies [23,24,25,26,27,28,29,30,31] was judged as follows: two studies (22.2%) had a low risk of bias, four studies (44.5%) had some concerns, and three studies (33.3%) had a high risk of bias (Figure 3). The main reason for the high overall risk of bias was primarily due to bias arising from missing outcome data. Detailed domain-specific assessments for the nine studies according to ROB 2.0 are provided in the Appendix A

## 4. Discussion

A previous study [19] compared four RCTs of a Qigong group to a no-treatment group and reported that Qigong is effective in alleviating stress. This systematic review found that Qigong has significant effects on stress reduction compared to the no-intervention group, stretching training exercises, and cognitive-behavioral therapy. These findings support the conclusions of the previous study [19] and suggest the potential for Qigong to serve as a replacement for existing mind–body therapies. Ng et al. [32] demonstrated that Qigong increases the counts of leukocytes and lymphocytes, cardiac output, early diastolic filling rate, late diastolic filling rate, forced vital capacity, and forced expiratory volume, while also having some effect in lowering total cholesterol, systolic blood pressure, diastolic blood pressure, and depression scores. They explained that these effects may be related to stress reduction through the nervous, endocrine, and immune systems. In this study, we aimed to verify Ng et al.’s explanation; we confirmed that Qigong has stress-reducing effects and found that it also positively impacts cortisol levels through biomarkers. This suggests that participating in a Qigong program under the supervision of a practitioner for a certain period can reduce perceived stress and influence changes in stress hormone (cortisol) levels. However, the limited number of studies included in this research and the heterogeneity of Qigong interventions limit the ability to draw definitive conclusions.

In the study by Oh et al. [33], the effects of Qigong on the immune and inflammatory systems were analyzed; they reported that while there is a significant effect on increasing immune cell counts, there is no significant effect on improving inflammation levels. In our study, plasma concentrations of IL-6 and IL-1β were assessed in only two studies [28,29]; one study demonstrated a significant effect, while the other showed no difference between the two groups. To confirm whether Qigong has a significant effect on increasing immune cell counts, it is necessary to examine indicators such as lymphocytes, natural killer cells, monocytes, CD4+ T cells, CD8+ T cells, and CD57+ T cells. Future RCTs of Qigong that include these indicators could provide stronger evidence for its effectiveness in enhancing the immune system, potentially extending its applications to cancer care.

In two studies [23,30], there was no significant effect of Qigong on pain improvement compared to the control group. However, a previous study [34] reported that Qigong significantly improves pain and stiffness. The studies included in this review did not show any effects on pain improvement because they used Qigong programs aimed at reducing stress. Regarding salivary biomarkers and complete blood counts, some studies reported positive effects, whereas others found no differences between the two groups. This suggests that the emotional improvement effects of Qigong therapy did not translate into physical changes. However, considering that some studies have shown significant effects in the Qigong group, further research is needed to verify whether extending the duration or increasing the frequency of treatment could improve salivary biomarkers and complete blood counts.

Qigong has been reported to have a positive effect on QoL. Job stress and emotional labor are directly related to physical and mental health and have been reported to affect QoL [35,36]. Modern people spend most of their time at work and suffer from job stress and emotional labor. If companies implement Qigong programs as part of employee stress management, they can positively impact their productivity and performance.

Two studies [23,28] reported no adverse effects, and one study [30] reported mild muscle soreness in six participants. As a non-pharmacological intervention, Qigong can provide physical and mental benefits without serious side effects. However, Guo et al. [37] reported that Qigong can cause side effects such as distension of the head, palpitation, shortness of breath, hypochondriac distension, muscular soreness or pain, neurasthenia, affective disorders, hallucinations, and paranoia. Standardized Qigong programs and training programs are necessary to minimize the potential adverse events caused by Qigong. Qigong practitioners should be well informed about the potential adverse events and run standardized Qigong programs.

The results of this review should be interpreted with caution due to the following limitations. This study conducted a systematic literature review of RCTs on Qigong therapy for stress by searching international and Korean databases, and ultimately selected nine studies. In countries such as China, Japan, and India, where traditional medicine is institutionalized, Qigong is used for health management and treatment [38,39,40]. However, this study did not include Chinese, Japanese, or Indian databases in the systematic literature review. It is necessary to supplement systematic reviews related to Qigong therapy for stress in future research by referring to the results of this study. In particular, it is necessary to conduct research that includes databases from major countries with traditional medicine practices, such as China, Japan, India, and Korea, through multinational studies. Second, due to the heterogeneity of the control group and the measurement tools used, as well as the statistical representations, only some studies related to the stress scale were included in the meta-analysis. For future meta-analyses, it will be necessary to use the same assessment tools for stress, depression, anxiety, and QoL, and provide statistical data post-treatment either in the main text or Appendix A. A meta-analysis is a statistical technique that quantitatively synthesizes the combined average effect size of the same intervention from individual clinical studies to draw conclusions. In the future, developing a standardized Qigong program through multinational studies could enable meta-analyses and contribute to health management in the general public and clinical fields. Third, the overall quality of all the RCTs, except for one study [30], was generally low. In particular, four studies had a dropout rate exceeding 20%, which could introduce bias in estimating the treatment effect. Fourth, training intensity using variables such as rate of perceived exertion or maximal heart rate was not reported in nine RCTs. This is a key factor that can influence the study outcomes, and it is recommended to include this in future Qigong RCTs for publication. Lastly, the studies included in this review did not account for the use of drugs, medical devices, or standard medical practices for stress management. In the nine RCTs included in this study, the experimental groups used Qigong intervention alone. Therefore, it was difficult to clarify the additional benefits of Qigong therapy alone or when combined with conventional treatment for stress reduction. In Europe, some countries consider Qigong as a preventive measure in health systems [41]. Specifically, in Germany, Qigong has become an essential component in both the prevention and supportive treatment of chronic diseases, with German family doctors emphasizing its role not only as a technique, but as a spiritual philosophy that significantly enhances its therapeutic value [42]. Future research is needed to clarify the relationship between Qigong therapy and conventional treatment in order to confirm its value as a complementary and alternative medicine.

Despite these limitations, this is the first systematic review of Qigong therapy for stress management, providing clinical evidence for managing stress in modern individuals.

## 5. Conclusions

Qigong has been used for thousands of years to optimize and restore the mind, body, and energy, and to enhance health and vitality through the regulation of the body, voice, breath, and mind. However, there is a growing trend of not recognizing practices, drugs, and medical devices that are not evidence-based, as legitimate medical practices. Our findings suggest that Qigong therapy is effective for managing stress, anxiety, and depression. However, further validation is required before it can be recommended as an evidence-based treatment in clinical settings. It will be necessary to conduct high-quality, multicenter, randomized controlled trials to strengthen the evidence for the use of Qigong in stress management.

## Figures and Tables

**Figure 1 healthcare-12-02342-f001:**
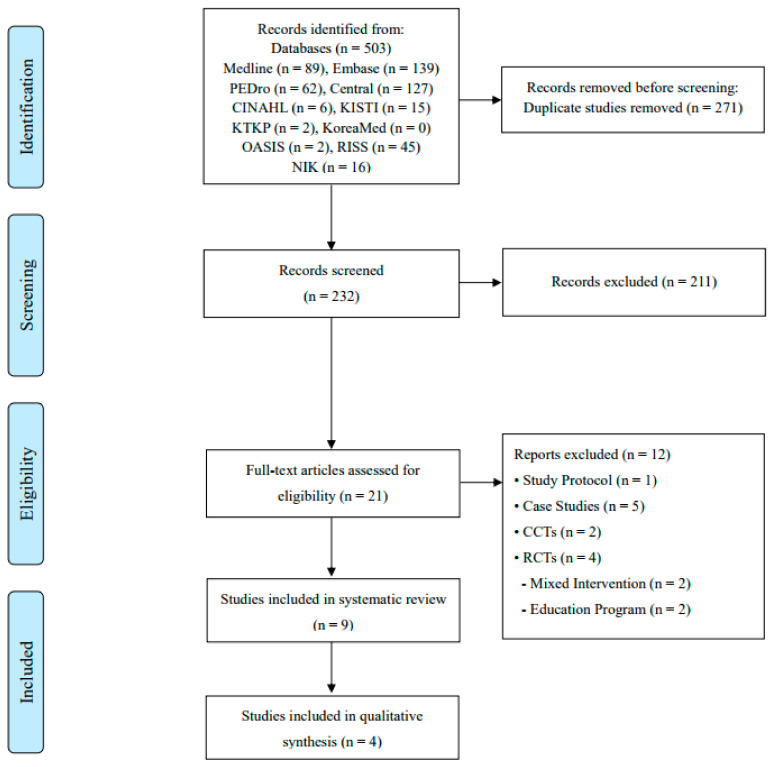
PRISMA study flow diagram. CCTs, controlled clinical trials; RCTs, randomized controlled trials.

**Figure 2 healthcare-12-02342-f002:**
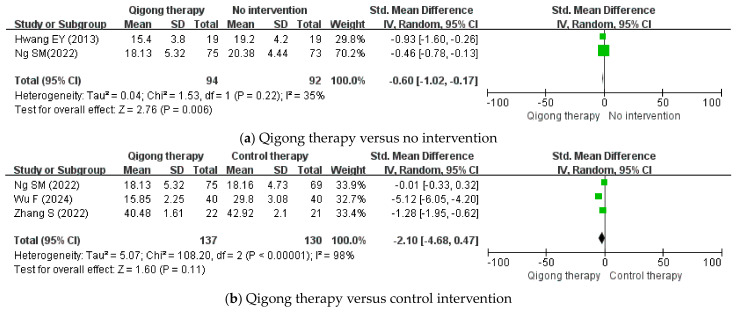
Meta-analysis of perceived stress scale for Qigong therapy. CI: Confidence interval [27,29,30,31].

**Figure 3 healthcare-12-02342-f003:**
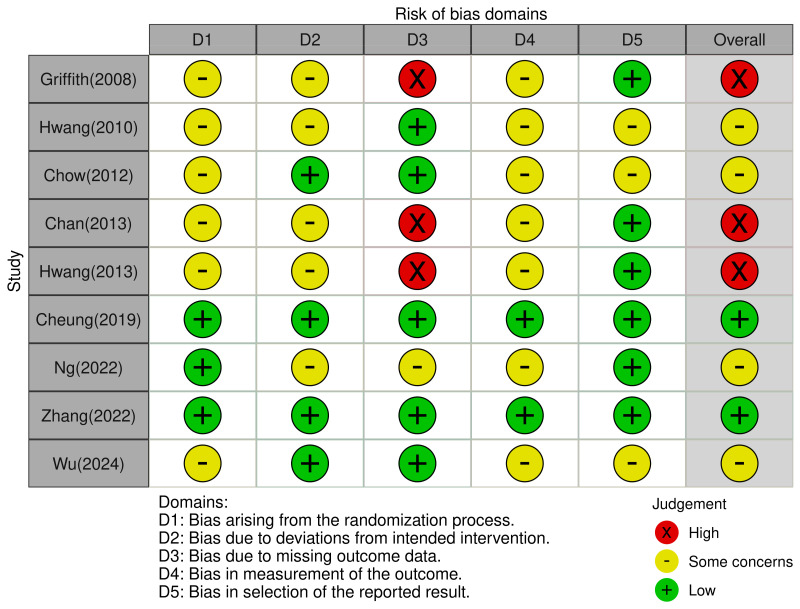
Risk of bias assessment [23,24,25,26,27,28,29,30,31].

**Table 1 healthcare-12-02342-t001:** Characteristics of the included studies.

First Author (Year) Country	Patient’s Disease, Gender (M/F), Age (Mean ± SD), Sample Size (Randomized/Analyzed)	Qigong Intervention	Control Intervention	Outcome Measurements	Main Result (Mean ± SD)
Griffith JM (2008) [23] United States	Hospital staff with stress, Gender: IG: 4/12, CG: 5/17 Age: IG: 52 ± 9, CG: 50 ± 10, Sample: 50/37	Qigong therapy (exercise), *n* = 16, total 12 sessions (twice a week for 6 weeks, 1 h for 1 session)	No intervention, *n* = 21	Primary outcome 1. Stress scale (PSS) Secondary outcomes 2. QoL scale (SF-36) (1) Physical Functioning (2) Role-Physical (3) Bodily Pain (4) General Health (5) Vitality (6) Social Functioning (7) Role-Emotional (8) Mental Health 3. Pain scale (pain score)	Change in measures from baseline 1. *p* = 0.02, (IG)–4.5 ± 6.6, (CG) 0.4 ± 4.9 2. (1) NS, (IG) 3.4 ± 14.2, (CG) 1.7 ± 7.5 (2) NS, (IG) 3.1 ± 35.2, (CG) 14.3 ± 35.0 (3) NS, (IG) –8.3 ± 24.1, (CG) 4.8 ± 14.6 (4) NS, (IG) 1.9 ± 18.4, (CG) 3.6 ± 15.8 (5) NS, (IG) 11.3 ± 19.2, (CG) 3.8 ± 14.9 (6) *p* = 0.05, (IG) 17.3 ± 24.5, (CG) 2.4 ± 16.7 (7) NS, (IG) 14.6 ± 24.1, (CG) 19.1 ± 34.3 (8) NS, (IG) –73.3 ± 17.3, (CG) 69.1 ± 24.5 3. NS, (IG) –5.6 ± 10.7, (CG) –0.3 ± 18.4
Hwang EY (2010) [24] South Korea	People with stress, Gender: 9/29 Age: 39.4 ± 11.4, Sample: 50/38	Qigong therapy (BQSRP), *n* = 19, total 56 sessions (twice a day for 4 weeks, 15 m for 1 session)	No intervention, *n* = 19	Primary outcome 1. Stress scale (PSS) Secondary outcomes 2. Anxiety scale (STAI) (1) State anxiety (2) Trait anxiety 3. QoL scale (WHOQOL-BREF) (1) Physical QoL (2) Psychological QoL (3) Social relation (4) Environment	Change in measures from baseline 1. (1) High HA group: *p* = 0.00, (IG) −6.8 ± 2.57, (CG) 0.67 ± 3.53 (2) Low group: *p* = 0.00, (IG) –5.54 ± 4.48, (CG) 1.33 ± 3.28 2. (1) -High HA group: *p* = 0.00, (IG) –8.60 ± 7.31, (CG) 2.83 ± 8.69 -Low group: *p* = 0.02, (IG) –8.23 ± 9.23, (CG) 1.78 ± 7.95 (2) -High HA group: *p* = 0.00, (IG) −6.1 ± 10.21, (CG) 4.75 ± 4.27 -Low group: *p* = 0.02, (IG) −5.38 ± 9.55, (CG) 3.44 ± 4.69 3. (1) -High HA group: *p* = 0.01, (IG) 1.97 ± 1.57, (CG) −0.76 ± 2.47 -Low group: NS, (IG) 1.29 ± 2.36, (CG) −0.31 ± 2.51 (2) -High HA group: *p* = 0.03, (IG) 1.85 ± 2.05, (CG) 0.00 ± 1.50 -Low group: NS, (IG) 1.72 ± 1.92, (CG) 0.30 ± 1.46 (3) -High HA group: *p* = 0.05, (IG) 1.04 ± 2.38, (CG) −1.00 ± 2.14 -Low group: *p* = 0.00, (IG) 1.44 ± 2.63, (CG) −1.93 ± 1.51 (4) -High HA group: NS, (IG) 1.78 ± 1.80, (CG) 0.46 ± 1.63 -Low group: NS, (IG) 1.50 ± 1.88, (CG) 0.61 ± 1.50
Chow YWY (2012) [25] China	Middle-aged adults with stress, Gender: 23/45 Age: 44.2 ± 11.03, Sample: 68/65	Qigong therapy (Chan Mi Gong Qigong exercises), *n* = 34, total 8 sessions (once a week for 8 weeks, 90 m for 1 session)	No intervention, *n* = 31	Primary outcome 1. Stress scale (DASS-stress) Secondary outcomes 2. Depression scale (DASS-depression) 3. Anxiety scale (DASS-anxiety) 4. QoL scale (ChQoL) 5. Salivary biomarker (cortisol level)	Measures after interventions 1. *p* = 0.019, (IG) 9.18 ± 4.95, (CG) 13.94 ± 5.92 2. NS, (IG) 2.47 ± 3.38, (CG) 6.65 ± 6.40 3. *p* = 0.034, (IG) 4.18 ± 3.24, (CG) 9.29 ± 7.02 4. *p* = 0.017, (IG) 70.62 ± 9.88, (CG) 57.71 ± 17.65 5. *p* < 0.001, (IG) 2088.80 ± 879.90, (CG) 2699.51 ± 1122.84
Chan ES (2013) [26] Brunei Darussalam	Students with stress, Gender: IG: 5/13, CG: 2/14 Age: IG: 19–28, CG:18–21, Sample: 46/34	Qigong therapy (exercise), *n* = 18, total 20 sessions (twice a week for 10 weeks, 1 h for 1 session)	No intervention, *n* = 16	Primary outcome 1. Stress scale (DASS-stress) Secondary outcomes 2. Depression scale (DASS-depression) 3. Anxiety scale (DASS-anxiety) 4. Self-assessed health scale 5. Salivary biomarkers (1) lgA (2) Cortisol level	Measures after interventions 1. n.r., *p* < 0.05 in (IG) 5 (range 0–18), NS in (CG) 8 (range 2–24) 2. n.r., *p* < 0.05 in (IG) 4 (range 0–14), NS in (CG) 2 (range 0–20) 3. n.r., *p* < 0.05 in (IG) 5 (range 0–20), NS in (CG) 6 (range 0–20) 4. n.r., NS in (IG) 0.18 (range 0.00–0.46), NS in (CG) 0.21 (range 0.00–0.43) 5. (1) n.r., *p* < 0.05 in (IG) 72.9 (range 36.6–205.8), NS in (CG) 67.4 (range 20.0–2223.0) (2) n.r., *p* < 0.05 in (IG) 3.3 (range 1.0–11.7), NS in (CG) 4.3 (range 1.9–7.7)
Hwang EY (2013) [27] South Korea	People with stress, Gender: 11/39 Age: IG: 41.48 ± 11.57, CG: 40.56 ± 11.75 Sample: 50/38	Qigong therapy (BQSRP), *n* = 19, total 16 sessions (4 times a week for 4 weeks, 2 h for 1 session)	No intervention, *n* = 19	Primary outcome 1. Stress scale (PSS) Secondary outcomes 2. Anxiety scale (STAI) (1) State anxiety (2) Trait anxiety 3. QoL scale (WHOQOL-BREF) (1) Physical QoL (2) Psychological QoL (3) Social Relation (4) Environment	Measures after interventions 1. *p* = 0.0006, (IG) 15.4 ± 3.8, (C) 19.2 ± 4.2 2. (1) *p* = 0.0028, (IG) 38.9 ± 10.0, (CG) 48.7 ± 10.6 (2) *p* < 0.0001, (IG) 41.0 ± 7.7, (CG) 49.8 ± 9.4 3. (1) *p* = 0.0151, (IG) 14.3 ± 1.9, (CG) 11.7 ± 3.8 (2) *p* = 0.0111, (IG) 12.7 ± 2.3, (CG) 11.3 ± 2.7 (3) *p* = 0.0140, (IG) 13.1 ± 1.8, (CG) 11.6 ± 2.8 (4) *p* = 0.0314, (IG) 12.7 ± 2.6, (CG) 12.0 ± 2.6
Cheung DST (2019) [28] China	Women with stress, Gender: 0/271 Age: IG: 42.0 ± 8.7, CG: 41.5 ± 9.3 Sample: 271/247	Qigong therapy (exercise), *n* = 120, total 28 sessions (twice a week for 1–6 weeks and once a week for 7–22 weeks, 30 m for 1 session)	No intervention, *n* = 127	Primary outcome 1. Stress scale (PSS) Secondary outcomes 2. Depression scale (BDI) 3. Biomarkers (Complete blood count) (1) Plasma concentrations of IL-6 (2) Plasma concentrations of TNF (3) Telomerase activity	Measures after interventions 1. *p* = 0.02, (IG) 17.76 (95% CI 16.72–18.81), (C) 19.57 (95% CI 18.55–20.60) 2. *p* = 0.009, (IG) 11.03 (95% CI 9.12–12.94), (C) 14.60 (95% CI 12.73–16.48) 3. (1) NS, (IG) 0.12 (95% CI–0.07–9.30), (C) 0.13 (95% CI −0.06–0.31) (2) NS, (IG) 1.35 (95% CI 1.01–1.70), (C) 1.29 (95% CI 0.95–1.63) (3) NS, (IG) 5.18 (95% CI 5.05–5.31), (C) 5.14 (95% CI 5.01–5.27)
Ng SM (2022) [29] China	People with stress, insomnia and depression, Gender: IG: 20/75, CG1: 20/73, CG2: 20/67 Age: IG: 55.97 ± 10.76, CG1: 54.59 ± 19.23, CG2: 55.91 ± 9.37 Sample: 281/217	(IG) Qigong therapy (exercise), *n* = 75, total 8 sessions (once a week for 8 weeks, 3 h for 1 session)	(CG1) No intervention, *n* = 73 (CG2) Integrative Body–Mind–Spirit, *n* = 69, total 8 sessions (once a week for 8 weeks, 3 h for 1 session)	Primary outcome 1. Stress scale (PSS) Secondary outcomes 2. Depression scale (CES-D) 3. Health scale (1) Physical symptoms (SSI) (a) SSI-pain (b) SSI-painless (2) Pittsburgh sleep quality index (3) Mind-body scale (a) Affliction (b) Equanimity 4. Complete blood count (1) Plasma concentrations of IL-6 (2) Plasma concentrations of IL-1β	Measures after interventions 1. (IG vs. CG1) *p* < 0.001, (IG) 18.13 ± 5.32, (CG1) 20.38 ± 4.44 (IG vs. CG2) NS, (IG) 18.13 ± 5.32, (CG2) 18.16 ± 4.73 2. (IG vs. CG1) *p* < 0.001, (IG) 16.77 ± 9.40, (CG1) 19.23 ± 7.75 (IG vs. CG2) NS, (IG) 16.77 ± 9.40, (CG2) 15.29 ± 8.12 3. (1) (a) (IG vs. CG1) *p* = 0.019, (IG) 15.35 ± 5.38, (CG1) 15.86 ± 5.81 (IG vs. CG2) NS, (IG) 15.35 ± 5.38, (CG2) 13.87 ± 5.23 (b) (IG vs. CG1) *p* < 0.001, (IG) 40.25 ± 12.06, (CG1) 42.91 ± 14.08 (IG vs. CG2) NS, (IG) 40.25 ± 12.06, (CG2) 38.05 ± 10.96 (2) (IG vs. CG1) *p* < 0.001, (IG) 12.37 ± 3.83, (CG1) 12.60 ± 3.12 (IG vs. CG2) NS, *p* < 0.001, (IG) 12.37 ± 3.83, (CG2) 11.11 ± 3.78 (3) (a) (IG vs. CG1) *p* < 0.001, (IG) 58.12 ± 22.33, (CG1) 63.94 ± 20.14 (IG vs. CG2) NS, (IG) 58.12 ± 22.33, (CG2) 56.78 ± 19.25 (b) (IG vs. CG1) *p* = 0.002, (IG) 94.35 ± 21.16, (CG1) 88.13 ± 19.18 (IG vs. CG2) NS, (IG) 94.35 ± 21.16, (CG2) 96.04 ± 20.98 4. (1) (IG vs. CG1) *p* = 0.015, (IG) 1.04 ± 0.56, (CG1) 1.30 ± 0.86 (IG vs. CG2) NS, (IG) 1.04 ± 0.56, (CG2) 1.05 ± 0.61 (2) (IG vs. CG1) *p* < 0.001, (IG) 0.05 ± 0.14, (CG1) 0.12 ± 0.27 (IG vs. CG2) NS, (IG) 0.05 ± 0.14, (CG2) 0.07 ± 0.15
Zhang S (2022) [30] China	Knee osteoarthritis patients with stress, Gender: IG: 4/21, CG: 9/16 Age: IG: 55.76 ± 8.37, CG: 53.40 ± 10.66, Sample: 50/43	Qigong therapy (Yijinjing Qigong), *n* = 22, total 24 sessions (twice a week for 12 weeks, 40 m for 1 session)	Stretching training exercise, *n* = 21, total 24 sessions (twice a week for 12 weeks, 40 m for 1 session)	Primary outcome 1. Stress scale (PSS) Secondary outcomes 2. Depression scale (BDI) 3. Health scale (1) Mental health scale (MCS) (2) Physical symptoms 4. Pain scale (1) VAS for pain (2) WOMAC scale for pain	Measures after interventions 1. *p* < 0.05, (IG) 40.48 ± 1.61, (CG) 42.92 ± 2.1 2. *p* < 0.01, (IG) 8.72 ± 1.42, (CG) 13.4 ± 2.1 3. (1) *p* < 0.001, (IG) 59.21 ± 9.04, (CG) 47.08 ± 7.65 (2) NS, (IG) 53.14 ± 6.64, (CG) 54.61 ± 4.64 4. (1) NS, (IG) 3.4 ± 0.91, (CG) 2.96 ± 0.97 (2) NS, (IG) 11.6 ± 1.44, (CG) 12.2 ± 1.38
Wu F (2024) [31] China	People with stress, anxiety, and depression, Gender: n.r. Age: IG: 20.50 ± 1.98, CG: 20.55 ± 1.44, Sample: 80/80	Qigong therapy (fitness), *n* = 40, total 45 sessions (5 times a week for 9 weeks, 1 h for 1 session)	Cognitive-behavioral therapy, *n* = 40, total 45 sessions (5 times a week for 9 weeks, 1 h for 1 session)	Primary outcome 1. Stress scale (PSS) Secondary outcomes 2. Depression scale (HAMD-24) 3. Anxiety scale (GAD-7)	Measures after interventions 1. *p* < 0.05, (IG) 15.85 ± 2.25, (CG) 29.80 ± 3.08 2. *p* < 0.01, (IG) 11.40 ± 3.82, (CG) 15.43 ± 3.80 3. *p* < 0.05, (IG) 7.84 ± 1.57, (CG) 9.85 ± 2.52

M: Male, F: Female, IG: Intervention Group, CG: Control Group, h: hour, PSS: Perceived Stress Scale, QoL: Quality of Life, NS: No Significant differences between groups, BQSRP: Brief Qigong-based Stress Reduction Program, STAI: State-Trait Anxiety Inventory, WHQoL-BREF: World Health Organization Quality-of-Life Scale, HA: Harm Avoidance, SD: Self-Directedness, DASS: Depression Anxiety Stress Scales, ChQoL: Chinese Quality of Life instrument, IgA: Immunoglobulin A, n.r.: not reported, BDI: Beck’s Depression Inventory, IL: Interleukin, m:munite, TNF: Tumor Necrosis Factor, CES-D: Center for Epidemiological Studies-Depression Scale, SSI: Somatic Symptom Disorder, MCS: Mental Component Score, VAS: Visual Analogue Scale, WOMAC scale: Western Ontario and McMaster Universities Osteoarthritis Index Scale, HAMD-24: Hamilton Rating Scale for Depression-24, SF-36: Short Form-36, GAD-7: Generalized Anxiety Disorder-7.

## Data Availability

The raw data supporting the conclusions of this article will be made available by the authors on request.

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
