# Peer review of "Qigong Therapy for Stress Management: A Systematic Review of Randomized Controlled Trials"

_healthcare, 2024, doi:10.3390/healthcare12232342_

Round 1
Reviewer 1 Report
Comments and Suggestions for Authors
Abstract: specify the resources used for the searches and the inclusion/exclusion criteria in relation to the population. Specify the method used for risk assessment and synthesis of results. In the results, more information on the main findings (including statistically relevant information) should be provided. It is recommended to be guided by the PRISMA for abstract extension.
Introduction: Information on methodology should not be included in the introduction (remove the last sentence on study design and on the use of PRISMA and move this information to the methodology). End the introduction with a description of the objective of the review.
Methods: I recommend changing 2.1 Protocol and registration to 2.1 Design. Start with the design of the review and include in this section the information on registration in PROSPERO and PRISMA.
In point 2.2, do not duplicate PubMed and Medline (they should leave only Medline, which is the database; PubMed is the search engine for searching Medline).
Specify the search dates in each of the databases and the search strategy in each database (it is not correct to describe in this section the dates up to which searches were carried out, as the time limit corresponds to inclusion criteria; transfer this information to inclusion/exclusion criteria). It is recommended to include a table or supplementary file as an annex with this information.
Since this is a review of the effectiveness of an intervention (Qigong), the research question must be identified with a PICO structure. It is important to define the intervention. (for this purpose improve the description of the information provided in points 2.3.1, 2.3.2, 2.3.3 and 2.3.4). It is important to identify the main outcome (¿STRESS?) on which results will be extracted from the included studies. Also possible secondary outcomes. The title of the manuscript indicates that the main outcome is stress.
Clarify inclusion and exclusion criteria in terms of population, study design, outcomes.
With regard to point 2.4, it should be specified how discrepancies in the screening process have been resolved (the same should be described for the screening process and risk of bias assessment).
With regard to data extraction, the scales used in the studies should not be included (this should be justified, as it seems that the authors are familiar with each and every scale used in the scientific literature). This should not be the case, as this information should correspond to the results of the review).
In point 2.6 there is an incomplete sentence (We evaluated the...) Revise this erratum. In this regard, the use of the first person plural should be corrected in the manuscript (check e.g. points 2.1, 2.2, 2.3.1 and following).
Information regarding meta-analysis should be improved in relation to the data used from the studies to be included (means, standardised means, standard deviation...). This aspect of the review is the one I consider to have the greatest inconsistency, as it indicates that meta-analyses were conducted, but these analyses do not appear in the results. There is also no justification for not performing meta-analyses due to the heterogeneity of the primary studies.
Results: In the results section, the flowchart must be shown immediately after it is cited in the text (do not cite Table 1 before this).
In the flowchart of Figure 1, the information regarding the number of records identified in each database should be improved (it is not correct to separate the identification phase into two boxes, and studies should not be referred to in this phase, only records—studies should only be indicated in the "included studies" phase). The number of records removed due to duplication is missing and should be included in the flowchart (it is recommended to consult the PRISMA model for adjustments). The title of Figure 1 should be simplified.
It is not correct to state that n=211 records were retrieved from the databases (this should be after duplicates are removed). The information in the text should be more precise regarding identification, records removed due to duplication, records screened by title and abstract, records screened in full text, and included studies.
In Table 1, the information should be improved: in the first column, identify the author and year (in addition to the citation). The "Title" column is not necessary, as this information is irrelevant. In fact, this table does not only include general characteristics but also reports all specific data. The "Main Results" column does not provide precise information on both the intervention and control groups (such as statistical data on means, standard deviations, p-values, effect sizes, etc.), which are necessary for the feasibility of a meta-analysis. The information in Table 2 should not be separated from Table 1, as it provides details about the characteristics of the interventions performed. Consider modifying Table 1 to a horizontal format so that Table 2 can be included within it.
The results are not well presented in the manuscript. Consider restructuring the information in a different way. For example, section 3.4 does not provide substantial information. Section 3.5 does not present results on the outcomes of the studies (which, in my opinion, should be anxiety, depression, stress, sleep, pain, etc.); instead, the scales used to assess the outcomes are described as outcomes.
Subheading 3.11 on risk of bias assessment does not lead to the meta-analysis. A meta-analysis of the RCTs is not conducted, nor is its absence justified due to the heterogeneity of the outcomes in the studies included in the review.
In Table 3, the acronyms (L, U, H) should be explained in a footnote at the bottom of the table.
Discussion: Review the use of acronyms throughout the manuscript. It is not appropriate to explain "RCT" in the discussion section if it has already been explained earlier. This aspect should be checked across the entire manuscript for other terms and acronyms as well.
Author Response
We appreciate the time given and efforts made by the editor and
referees in reviewing this paper. We have addressed all issues indicated
in the review report in a point by point manner, and changed those parts
in red. We hope that the revised paper will be able to meet the journal
publication requirements.
Once again, we very much appreciate you taking the time to comment
on our manuscript.
Please see the attachment.

Reviewer 2 Report
Comments and Suggestions for Authors
General Comments
Thank you for the opportunity to review your paper. Overall, the topic is engaging and aligns well with the scope of the journal. However, there are several recommendations for improvement that I believe should be taken into account.
Specific Comments
L23-24: In some instances, you use "Qigong," while in others, "qigong." Please ensure consistency throughout the manuscript.
L31-34: The conclusion is too general and requires greater precision.
L35: Several of your keywords are already included in the title. If possible, please provide different keywords to enhance the visibility of your review.
L86. The authors state that they adhered to PRISMA guidelines; however, several critical details are missing. For instance, why is there no mention of the selection process in the methodology? Why was the quality of the publications not assessed? Was a sensitivity analysis conducted, and if so, what were the results? Furthermore, it is essential to include the PRISMA checklist as supplementary material to ensure full transparency and compliance with the guidelines.
L94-95. Considering the limited number of studies included and the fact that nearly a year has elapsed since the search was conducted, an update is required.
L106-109. You should provide more details in the inclusion and exclusion criteria. Were there any language restrictions?
L107. Mentioned "in this meta-analysis," yet this information is not reflected in the title or abstract of your study.
L113. Replace "sex" with "gender" to adopt more inclusive language.
L146-155. You mention having conducted a data analysis with RevMan; however, these data are not presented. Please include this information in the results section if the meta-analysis was conducted. If not, remove section 2.6 Data Analyses to avoid confusing the reader.
L188: The description of the sample is also imprecise. Do the studies provide information about the participants' medication use, drop-outs, adherence, sports experience, age or gender? This information is crucial for interpreting the results and identifying potential gaps in the knowledge.
L196. How many studies reported the intensity of the interventions? How was the intensity measured? What was the duration of the training sessions? These details are essential to provide a comprehensive understanding of the interventions and should be addressed in the results section.
L299. The discussion is too superficial, as it merely describes the results. A more thorough discussion and comparison with other therapies, previous research, etc., is necessary. This is a critical point.
L301. Abbreviations should only be explained the first time they are used, for example, RCTs. Please review the entire manuscript to ensure this change is made appropriately.
Author Response

(The authors gave the same response as above.)

Reviewer 3 Report
Comments and Suggestions for Authors
There are references and studies about the officially recognition of Qigong as complementary medicine also in Europe?
How were drugs, medical devices and legitimate medical practices due to the large amount of pathology, taken into account on this study?
Comments on the Quality of English LanguageMinor revisions on scientific english
Author Response

(The authors gave the same response as above.)

Round 2
Reviewer 1 Report
Comments and Suggestions for Authors
Dear authors, thank you very much for the implementations made in the manuscript.
I must point out some issues for improvement:
Generally speaking, the use of the first person plural in the abstract and the whole manuscript should be avoided (revise and replace the ‘We’ where appropriate).
Reference [30] should be deleted from the abstract.
Methods: The research question with a PICO structure must appear earlier in the methodology. Move this information to the design sub-heading; and specify each part of the PICO specifically in your research question.
Regarding to the inclusion and exclusion criteria for type of comparison, I consider that this sub-heading should be type of design (only RCTs were included). In fact, inclusion/exclusion criteria should be distinguished for: research study designs, participants, interventions.
Exclusion criteria should not be intermingled with inclusion criteria. State all inclusion criteria first and then all exclusion criteria.
The criterion type of outcomes does not, in my opinion, correspond to inclusion criteria (should be moved to another sub-heading on definition of study variables (do not describe them in a list, better to write them in the text).
In my opinion, the information on type of intervention should also be transferred to the sub-heading on study variables, where the outcomes (primary, secondary, additional) and their characteristics should be described.
The following studies were considered for inclusion: (1) full text articles, (2) English and Korean language papers, (3) peer-reviewed original research, and (4) RCT design. Exclusion criteria were (1) conference abstracts, letters, and comments, (2) non-English language articles, (3) non-RCTs among clinical studies (e.g. case reports, case series, and case-controlled trials), (4) animal in vivo or in vitro experimental studies, and (5) reviews, qualitative studies, surveys, and study protocols.
Review the wording of the following paragraph to make it more logical. Do not duplicate irrelevant information (e.g. it is irrelevant to say that peer reviewed original research is included as RCTs are included). Also, it seems logical to exclude non RCTs studies).
The ROB tool must be correctly cited as ROB 2 tool. Add the appropriate tool citation and reference.
The following sentence is not correct: Meta-analysis was not undertaken due to the heterogeneity of Qigong interventions and differences in data presentation in the included studies [22].
The methodology should explain the process for conducting the meta-analysis. In the results, the feasibility of the meta-analysis will be verified or not. But it is not correct to indicate the impossibility of performing the meta-analysis because different instruments have been used. The meta-analysis should be performed on the outcome ‘stress’ for the measurements of the mean and standard deviation of each study. The fact that different instruments have been used may be a cause of high heterogeneity, but if statistical data are available, the meta-analysis should be undertaken with these limitations in the results.
In this respect, although the content of table 1 has been substantially improved, it should be possible to identify more clearly in the last column to which outcome each statistical value corresponds. It is important that this information clearly distinguishes the mean and standard deviation values (as well as statistical significance and effect size (where applicable)) for both the control and intervention groups. The CG and IG should be clearly identified.
Results: In the flow chart they should not describe in the last box ‘studies included in the qualitative synthesis’. It is understood that n = 9 studies included in the review.
Figure 1 should be placed immediately after citing it in the text. Likewise for Table 1.
The results of the socio-demographic characteristics (3.2 participants) shall be described before the statistical results.
In table 1 some information should be corrected (for the control group it is agreed to use CG, instead of C; for the intervention group (IG) instead of E). In my opinion, some studies (e.g. Griffith, 2008 [23]; Chow, 2012 [25]; Zang, 2022 [30]) lack statistical data on the standard deviations of the results. I believe that some acronyms should be explained at the bottom of the table: M (mean) vs M (male)?; NS, n.r..
list in the table in order of appearance each of the acronyms explained at the foot of the table.
Table 2 is missing.
It is possible that some of the results do not simply describe the information in the results and duplicate information that is in the table. Some of this information in the text corresponds to the discussion of the studies.
The information on risk of bias assessment is provided, but I believe that this process should lead to meta-analysis of the results. It would be appropriate to also provide information on the degree of recommendation of the results for each outcome with GRADE.
In my opinion, meta-analysis is possible, at least for the main outcome (although it would also be possible for secondary outcomes) since there are statistical data that allow it. I consider that the following statement is not correct: "Second, a meta-analysis could not be performed due to the heterogeneity among the interventions and the lack of results data".
Author Response
Thank you for reviewer's valuable comments. We appreciate the time given and efforts made in reviewing this paper. We have addressed all issues indicated in the review report in a point by point manner, and changed those parts in red. Please see the attachment.

Reviewer 2 Report
Comments and Suggestions for Authors
I commend the authors for their efforts to enhance their work and appreciate their thoughtful consideration of my comments. I would kindly recommend a review of a few minor aspects before proceeding with the submission of their study for publication.
Reply 4: According to the comments, we have changed the keywords as
follows (line 43-44).
- Qigong; Qigong therapy, Stress; Stress reduction, Anxiety, Depression,
Quality of life
Comment 4: Remove “Qigong therapy”. It is redundant and already appears in the title
Reply 5: 3. The “2.3. Inclusion and Exclusion” section has been reorganized
according to the PICO framework (line 116-153). Outcomes have been
distinguished between primary and secondary outcomes.
Comment 5: Please present the inclusion/exclusion criteria as a PICOS Table to enhance the readability of this section.
Reply 12: We have included details on the intensity of the interventions (total
sessions, frequency per week, and duration of each session).
Comment 12: This information (total sessions, frequency per week, and duration of each session) does not address the training intensity. If no study references training intensity using variables such as RPE, %HRmax, or similar, please indicate this clearly.
Author Response

(The authors gave the same response as above.)
